# Proteolytic maturation of $\alpha_2\delta$ controls the probability of synaptic vesicular release

**Laurent Ferron\*, Ivan Kadurin, Annette C Dolphin**

Department of Neuroscience, Physiology and Pharmacology, University College London, London, United Kingdom

**Abstract** Auxiliary $\alpha_2\delta$ subunits are important proteins for trafficking of voltage-gated calcium channels ($Ca_V$) at the active zones of synapses. We have previously shown that the post-translational proteolytic cleavage of $\alpha_2\delta$ is essential for their modulatory effects on the trafficking of N-type ($Ca_V2.2$) calcium channels (Kadurin et al., 2016). We extend these results here by showing that the probability of presynaptic vesicular release is reduced when an uncleaved $\alpha_2\delta$ is expressed in rat neurons and that this inhibitory effect is reversed when cleavage of $\alpha_2\delta$ is restored. We also show that asynchronous release is influenced by the maturation of $\alpha_2\delta-1$, highlighting the role of $Ca_V$ channels in this component of vesicular release. We present additional evidence that $Ca_V2.2$ co-immunoprecipitates preferentially with cleaved wild-type $\alpha_2\delta$. Our data indicate that the proteolytic maturation increases the association of $\alpha_2\delta-1$ with $Ca_V$ channel complex and is essential for its function on synaptic release.

DOI: https://doi.org/10.7554/eLife.37507.001

## Introduction

Among the three families of $Ca_V$ channels ($Ca_V1$, $Ca_V2$ and $Ca_V3$), the $Ca_V2$ family and more specifically $Ca_V2.1$ and $Ca_V2.2$ channels (generating P/Q and N-type currents, respectively) are particularly important for synaptic transmission in central and peripheral nervous systems (*Dolphin, 2012*). $Ca_V2.1$ and $Ca_V2.2$ are targeted to presynaptic terminals where they are responsible for triggering vesicular release (*Catterall and Few, 2008*; *Zamponi et al., 2015*). $Ca_V$s are formed of several subunits: the $\alpha_1$ subunit, that constitutes the $Ca^{2+}$ selective pore and the voltage sensor, and auxiliary subunits $\beta$ (cytoplasmic) and $\alpha_2\delta$ (extracellular) (*Flockerzi et al., 1986*; *Liu et al., 1996*; *Takahashi and Catterall, 1987*; *Witcher et al., 1993*). Four genes coding for $\alpha_2\delta$ subunits have been identified (*Dolphin, 2012*). They are translated into a single pre-protein $\alpha_2\delta$ and post-translationally cleaved into $\alpha_2$ and $\delta$ peptides, which remain attached by di-sulfide bonds (*Dolphin, 2012*). In $\alpha_2\delta-1$ and $-2$, $\alpha_2$ contains a perfect metal ion adhesion site (MIDAS) motif essential for the interaction with $\alpha_1$ subunit (*Cantí et al., 2005*; *Hendrich et al., 2008*) and $\delta$ which is glycophosphatidyli-nositol (GPI) anchored to the plasma membrane (*Davies et al., 2010*). The structure of the $Ca_V1.1$ channel complex has been recently determined using cryo-electron microscopy and has identified binding domains between $Ca_V1.1$ and $\alpha_2\delta-1$ including the interaction of the $\alpha_2\delta$ MIDAS motif with loop I of the first repeated domain of $Ca_V1.1$ (*Wu et al., 2016*). Site-directed mutagenesis studies have confirmed a functional interaction between $\alpha_2\delta-1$ and the first extracellular loop of $Ca_V1.2$ (*Bourdin et al., 2017*) and $Ca_V2.2$ channels (unpublished results).

$\alpha_2\delta$ subunits are important for the trafficking of $\alpha_1$ subunits and their function, and they are also key proteins for synaptic function and synaptogenesis (*Dickman et al., 2008*; *Eroglu et al., 2009*; *Hoppa et al., 2012*; *Saheki and Bargmann, 2009*; *Zamponi et al., 2015*). We have recently shown that the proteolytic maturation of $\alpha_2\delta-1$ into disulfide-linked polypeptides $\alpha_2$ and $\delta$ is an essential post-translational step enabling its modulatory effect on the activation and trafficking of N-type calcium channels in neurons (*Kadurin et al., 2016*). Indeed, we show that uncleaved $\alpha_2\delta-1$ inhibits

**\*For correspondence:**
l.ferron@ucl.ac.uk

**Competing interests:** The authors declare that no competing interests exist.

presynaptic calcium transient-triggered action potential (AP) in hippocampal neurons and that this effect is reversed by the cleavage of $\alpha_2\delta-1$.

Here, we investigate the impact of the proteolytic maturation of $\alpha_2\delta-1$ on synaptic release. We used optical tools to measure vesicular release parameters (*Ariel and Ryan, 2010*; *Hoppa et al., 2012*). Our data show that an uncleaved $\alpha_2\delta-1$ reduces the probability of release in response to a single action potential, and also affects asynchronous release. These effects on presynaptic vesicular release are reversed when the cleavage of $\alpha_2\delta-1$ is restored. We provide additional evidence that cleaved $\alpha_2\delta-1$ interacts more than the uncleaved form with the Ca$_V$2.2 channel pore-forming subunit. Our data indicate that the proteolytic maturation of $\alpha_2\delta-1$ is important for its association with the Ca$_V$ channel complex and its function on synaptic release.

## Results

$\alpha_2\delta$ subunits play a crucial role in the trafficking of fully functional calcium channels to the plasma membrane and to presynaptic terminals (*Dolphin, 2012*). In order to determine the physiological impact of proteolytic maturation of $\alpha_2\delta-1$, we used the cleavage site mutant $\alpha_2(3C)\delta-1$ (*Kadurin et al., 2016*), which is resistant to endogenous proteolysis between $\alpha_2$ and $\delta$, to assess the effect of controlled cleavage by exogenous 3C-protease on vesicular release from presynaptic terminals, using the optical reporter vGlut-pHluorin. Transfected hippocampal neurons were identified by mCherry expression (*Figure 1a*). Neurons were subsequently stimulated (100 AP at 10 Hz), and fluorescence of vGlut-pHluorin was monitored to identify functional boutons (*Figure 1a*). We first examined the effect of expression of $\alpha_2(3C)\delta-1$ on synaptic release properties by measuring single AP-

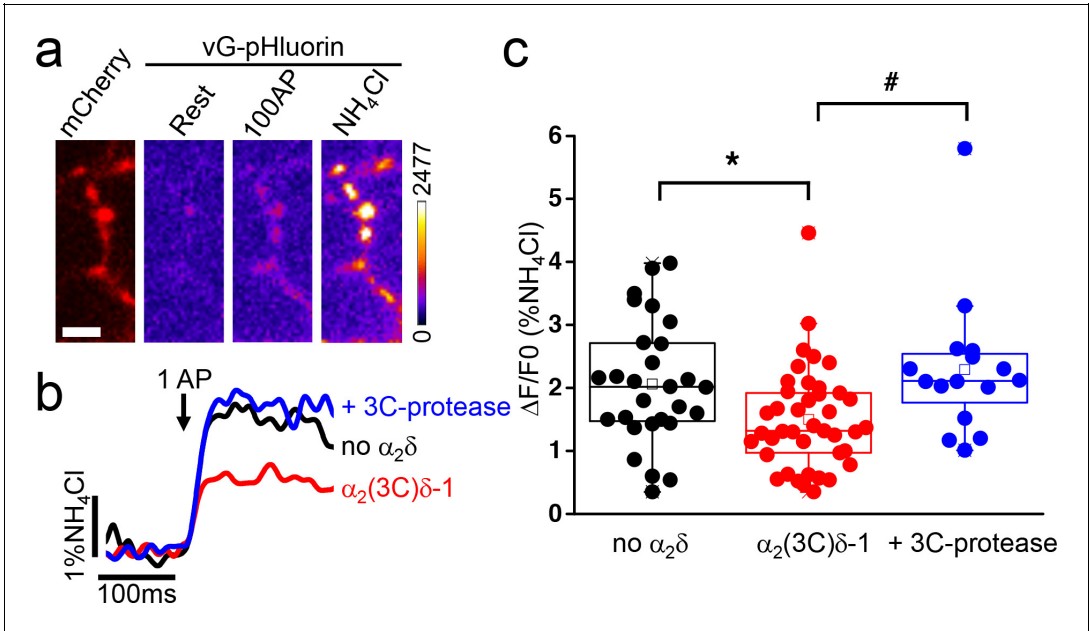

**Figure 1.** Effect of proteolytic cleavage of $\alpha_2(3C)\delta-1$ on vesicular release in presynaptic terminals of hippocampal neurons. (a) Fluorescence changes in presynaptic terminals of hippocampal neurons expressing vGlut-pHluorin (vG-pHluorin) in response to electrical stimulation. Left panel, mCherry-positive boutons. Three right panels, vG-pHluorin fluorescence: at rest (left), after 100 AP at 10 Hz (middle) and after a brief application of NH$_4$Cl (right). Scale bar 5 μm. The pseudocolour scale is shown with the last panel. (b) Representative vG-pHluorin responses to a single AP (10–12 trial average, 25 to 50 boutons) from presynaptic terminals of neurons co-transfected with empty vector (black trace), $\alpha_2(3C)\delta-1$ (red trace) or $\alpha_2(3C)\delta-1$ + 3C-protease (blue trace). Arrow indicates stimulation with one AP. (c) vG-pHluorin fluorescence changes (expressed as % of NH$_4$Cl response) in response to 1 AP from boutons co-transfected with empty vector (black), $\alpha_2(3C)\delta-1$ (red) or $\alpha_2(3C)\delta-1$ + 3C-protease (blue) (n = 28, 41 and 16 independent experiments, respectively). Box and whiskers plots; *p=0.044 and #p=0.014, one way ANOVA and Bonferroni post-hoc test.
DOI: https://doi.org/10.7554/eLife.37507.002

evoked exocytosis (*Figure 1b*). Single AP stimulations were repeated 10 to 12 times with a 45 s rest between each trial. Signals from each bouton were averaged and normalized to the fluorescence value obtained by rapid alkalinization of the entire labeled vesicle pool using $NH_4Cl$ (*Figure 1a and b*). Overexpression of uncleaved $\alpha_2(3C)\delta-1$ induced a decrease of 29 ± 6% in exocytosis compared to the control empty vector condition (n = 28 and 41, respectively; p=0.04) (*Figure 1c*). Conversely, inducing controlled cleavage of $\alpha_2(3C)\delta-1$ by co-expressing 3C-protease resulted in an increase of 53 ± 18% in exocytosis compared to $\alpha_2(3C)\delta-1$ alone (n = 41 and 16, respectively; p=0.014), thus completely reversing the inhibitory effect of uncleaved $\alpha_2(3C)\delta-1$ (*Figure 1b and c*).

Synaptic vesicle exocytosis properties are determined by the number of vesicles available for rapid release (the readily-releasable pool - RRP) and the probability (*Pv*) that a vesicle in the RRP will undergo fusion in response to a single AP stimulus (*Schneggenburger et al., 2002*). RRP can be determined using a high frequency stimulation (*Ariel and Ryan, 2010*; *Ariel et al., 2012*). During a 20 AP stimulus at 100 Hz, the fluorescence of vGlut-pHluorin in presynaptic terminals rapidly increases and reaches a plateau phase corresponding to the RRP (*Figure 2a*). The averaged response, obtained from 5 to 6 trials with a 5 min rest between each trial, were normalized to the size of the total presynaptic pool obtained with $NH_4Cl$ application (*Figure 2a*). To examine whether proteolytic maturation of $\alpha_2\delta-1$ affects the size of the RRP, we imaged neurons transfected with either empty vector (*Figure 2a*) or $\alpha_2(3C)\delta-1$ (*Figure 2b*) or $\alpha_2(3C)\delta-1$ together with 3C-protease (*Figure 2c*) and compared the size of the RRP. As summarized in *Figure 2d*, no difference was recorded between the three conditions (empty vector, $\alpha_2(3C)\delta-1$ and $\alpha_2(3C)\delta-1$ with 3C-protease: 6.9 ± 0.4, 6.2 ± 0.3 and 6.4 ± 0.5% of total pool, n = 22, 16 and 19, respectively, p=0.78), indicating that proteolytic maturation of $\alpha_2\delta-1$ affects the *Pv*, rather than the size of the RRP.

After the plateau phase corresponding to the RRP, an additional increase in fluorescence takes place during the stimulation, and continues for more than 500 ms after the end of the stimulus before reaching a stationary phase (*Figure 3a*). It was proposed that this secondary increase in fluorescence results from a combination of RRP refilling and slow decay of the elevated intracellular $Ca^{2+}$ concentration (*Ariel and Ryan, 2010*). This late increase in fluorescence occurs at lower rate than the initial increase and represents post-stimulus exocytosis. Overexpression of uncleaved $\alpha_2(3C)\delta-1$ induced a decrease of about 30% in this phase of exocytosis compared to control empty vector condition (n = 22 and 31, respectively; p<0.001) (*Figure 3b–c*). This reduction of delayed exocytosis is completely prevented by the co-expression of $\alpha_2(3C)\delta-1$ with 3C-protease (*Figure 3a–b*).

We then wished to determine whether the results obtained on presynaptic release were due to differential interaction of cleaved and uncleaved $\alpha_2\delta$ with the $\alpha1$ subunit. We have previously shown that transient expression of $\alpha_2\delta-1$ in cell lines results in only a partial cleavage of wild type $\alpha_2\delta-1$, such that a mixture of cleaved and uncleaved $\alpha_2\delta$ protein appears in the whole cell lysate (WCL) (*Kadurin et al., 2012*, *2016*). We performed co-immunoprecipitation of wild type $\alpha_2\delta-1$ with $Ca_V2.2$ from tsA-201 cell WCL and found that the percentage of cleaved $\alpha_2\delta-1$ in the co-immunoprecipitated fractions is ~4 fold higher than the percentage of cleaved $\alpha_2\delta-1$ in the input WCL (from 10.0 ± 0.6% to 39.2 ± 1.6% in WCL and co-immunoprecipitated fractions, respectively, n = 3) (*Figure 4*), suggesting stronger association of mature cleaved $\alpha_2\delta-1$ with the $Ca_V$ pore-forming subunit.

## Discussion

$Ca_V2$ channels are important for synaptic transmission and their targeting to the active zone is tightly regulated (*Catterall and Few, 2008*; *Simms and Zamponi, 2014*). $\alpha_2\delta$ subunits have been shown to control the trafficking of $Ca_V2$ to presynaptic terminals (*Hoppa et al., 2012*). $\alpha_2\delta$ subunits are post-translationally proteolysed, and this process is key for their regulatory action on $Ca_V2$ channels (*Kadurin et al., 2016*). Here, we show that the post-translational proteolytic maturation of $\alpha_2\delta-1$ is also essential for these proteins to fulfil their regulatory function on vesicular release in presynaptic terminals of hippocampal neurons in culture. Interestingly, we show that both synchronous and asynchronous releases are affected, both release mechanisms being highly dependent on $Ca^{2+}$ influx through $Ca_V2$ channels.

Vesicular release is characterized by two key presynaptic parameters: the RRP and *Pv* (*Ariel and Ryan, 2012*; *Schneggenburger et al., 2002*). A previous study has shown that over-expression of $\alpha_2\delta$ subunits and knock-down of endogenous $\alpha_2\delta$ increased and decreased *Pv*, respectively (*Hoppa et al., 2012*). In good agreement with this, our data show that uncleaved $\alpha_2\delta-1$ ($\alpha_2(3C)\delta-1$)

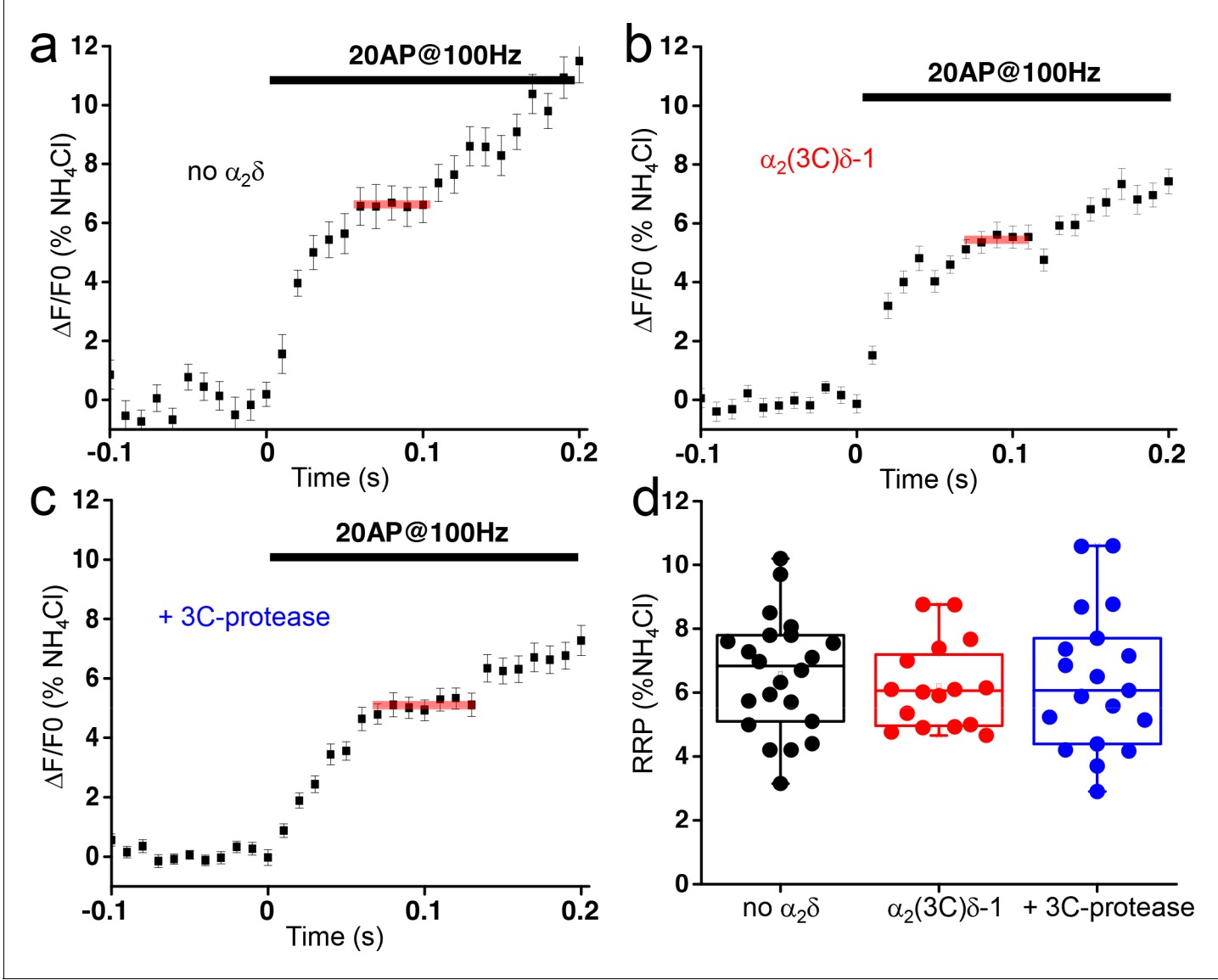

**Figure 2.** The proteolytic cleavage of $\alpha_2(3C)\delta$-1 does not affect the readily releasable pool (RRP) in presynaptic terminals of hippocampal neurons. (a–c) vG-pHluorin responses (mean ± SEM) to 20 AP at 100 Hz (5–6 trial average, 25 to 50 boutons) from presynaptic terminals of neurons co-transfected with empty vector (a), $\alpha_2(3C)\delta$-1 (b) or $\alpha_2(3C)\delta$-1 + 3C-protease (c). Horizontal red lines identify RRPs. (d) Average RRP (expressed as % of $NH_4Cl$ response) from boutons co-transfected with empty vector (black), $\alpha_2(3C)\delta$-1 (red) or $\alpha_2(3C)\delta$-1 + 3C-protease (blue) (n = 22, 16 and 19 independent experiments, respectively, p=0.78). Box and whiskers plots; one way ANOVA and Bonferroni post-hoc test.
DOI: https://doi.org/10.7554/eLife.37507.003

reduces $Pv$, and co-expression of the 3C-protease restores the control $Pv$. Interestingly, $Pv$ is modulated by the number of $Ca_V2$ channels in each active zone (**Ermolyuk et al., 2012**) and we have previously shown that uncleaved $\alpha_2\delta$ subunits reduced the amplitude of calcium transients triggered by a single AP stimulation, by interfering with the trafficking of $Ca_V2$ channels (**Kadurin et al., 2016**). The active zone proteins Rab-3 interacting molecules (RIMs) and Munc-13, critical in the orchestration of synaptic vesicular release, have been shown to control the targeting of $Ca_V2$ channels within presynaptic terminals (**de Jong et al., 2018**; **Südhof, 2012**). These active zone proteins have also been shown to control the size of the RRP (**Augustin et al., 1999**; **Calloway et al., 2015**; **Deng et al., 2011**; **Kaeser et al., 2011**). The RRP is defined as a small fraction of vesicles in a presynaptic terminal that is available for immediate release with a brief stimulus train, and thus likely to equate to docked vesicles identified by electron microscopy (**Ariel and Ryan, 2012**; **Rizzoli and**

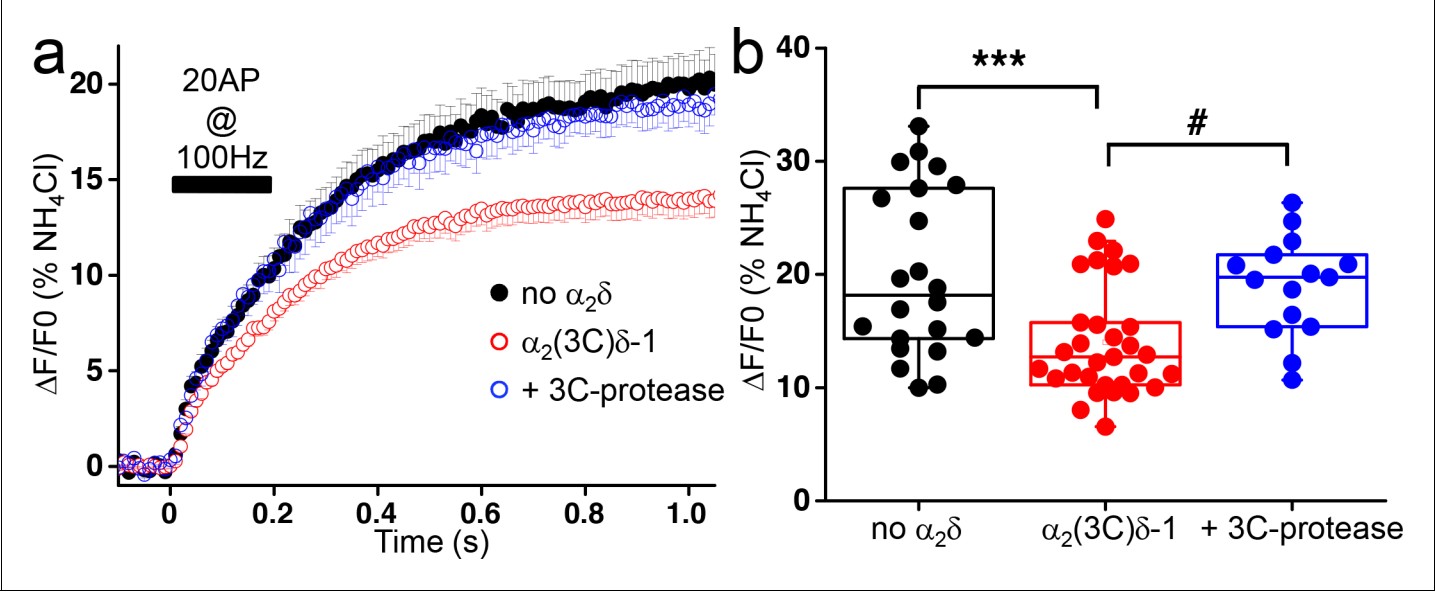

**Figure 3.** Effect of the proteolytic cleavage of $\alpha_2(3C)\delta$-1 on delayed vesicular release in presynaptic terminals of hippocampal neurons. (**a**) Average vG-pHluorin responses (mean ± SEM) to 20 APs at 100 Hz (5–6 trial average, 25 to 50 boutons) from presynaptic terminals of neurons co-transfected with empty vector (black), $\alpha_2(3C)\delta$-1 (red) or $\alpha_2(3C)\delta$-1 + 3C-protease (blue). The black bar indicates the stimulation period (20 AP at 100 Hz). (**b**) Average delayed vesicular release (expressed as % of NH4Cl response) measured 1 s after the beginning of the stimulation from boutons co-transfected with empty vector (black), $\alpha_2(3C)\delta$-1 (red) or $\alpha_2(3C)\delta$-1 + 3C-protease (blue) (n = 22, 31 and 15 independent experiments, respectively). Box and whiskers plots with superimposed individual experiments; ***p<0.001 and # p=0.021, one way ANOVA and Bonferroni post-hoc test.
DOI: https://doi.org/10.7554/eLife.37507.004

*Betz, 2005*; *Schneggenburger et al., 2002*). Experimental methods used to estimate the size of the RRP have been recently reviewed and consist of two electrophysiological methods (post synaptic current recordings and presynaptic membrane capacitance measurements) and one optical method (*Kaeser and Regehr, 2017*). Here, we used the optical technique that was developed by *Ariel and Ryan, (2010)*. This high-time resolution optical method measures exocytosis by detecting fluorescence from pHluorin tagged vGlut-1 (*Voglmaier et al., 2006*) associated with vesicle fusion. The high frequency stimulation protocol (20 APs at 100 Hz) induces a rapid rise in fluorescence followed by a plateau corresponding to a state during which all the vesicles in the RRP have fused with the membrane. The size of the RRP we describe here, which is determined by the amplitude of the fluorescence of the plateau (6–7% of the total pool of vesicles) is in good agreement with previously described values of RRP in neonatal rodent hippocampal neuron synapses (*Ariel and Ryan, 2010*; *Fernández-Alfonso and Ryan, 2006*; *Rizzoli and Betz, 2005*). A previous study has shown that wild-type $\alpha_2\delta$ subunits have no effect on the size of the RRP (*Hoppa et al., 2012*). Consistent with that study, our data show that uncleaved $\alpha_2\delta-1$ does not affect the size of the RRP indicating that, unlike RIMs and Munc13, $\alpha_2\delta-1$ does not have the same dual function on synaptic vesicular release.

There are two potential mechanisms to account for the reduction in *Pv* by $\alpha_2(3C)\delta-1$. It is likely that $\alpha_2(3C)\delta-1$ reduces the trafficking of endogenous Ca$_V$2 channels into active zones, as we showed for exogenously expressed Ca$_V$2.2 (*Kadurin et al., 2016*). However, $\alpha_2(3C)\delta-1$ can also traffic alone into presynaptic terminals (*Kadurin et al., 2016*), where it could then displace the endogenous $\alpha_2\delta$ interacting with channels in active zones, thus forming non-functional channels. The finding here that uncleaved $\alpha_2\delta$ interacts less than cleaved $\alpha_2\delta$ with Ca$_V$2.2 may indicate that the former mechanism is more likely.

Several reports have also described a role for $\alpha_2\delta$ subunits in synaptogenesis, independently from their role as a Ca$_V$ auxiliary subunit (*Dickman et al., 2008*; *Eroglu et al., 2009*; *Kurshan et al., 2009*). $\alpha_2\delta$ subunits are extracellular proteins anchored to the plasma membrane via a GPI moiety (*Davies et al., 2010*) which makes them potentially good candidates to interact with extracellular ligands such as thrombospondins, low density lipoprotein receptor-related protein and $\alpha$-neurexin (*Eroglu et al., 2009*; *Kadurin et al., 2017*; *Tong et al., 2017*). Although a direct interaction between

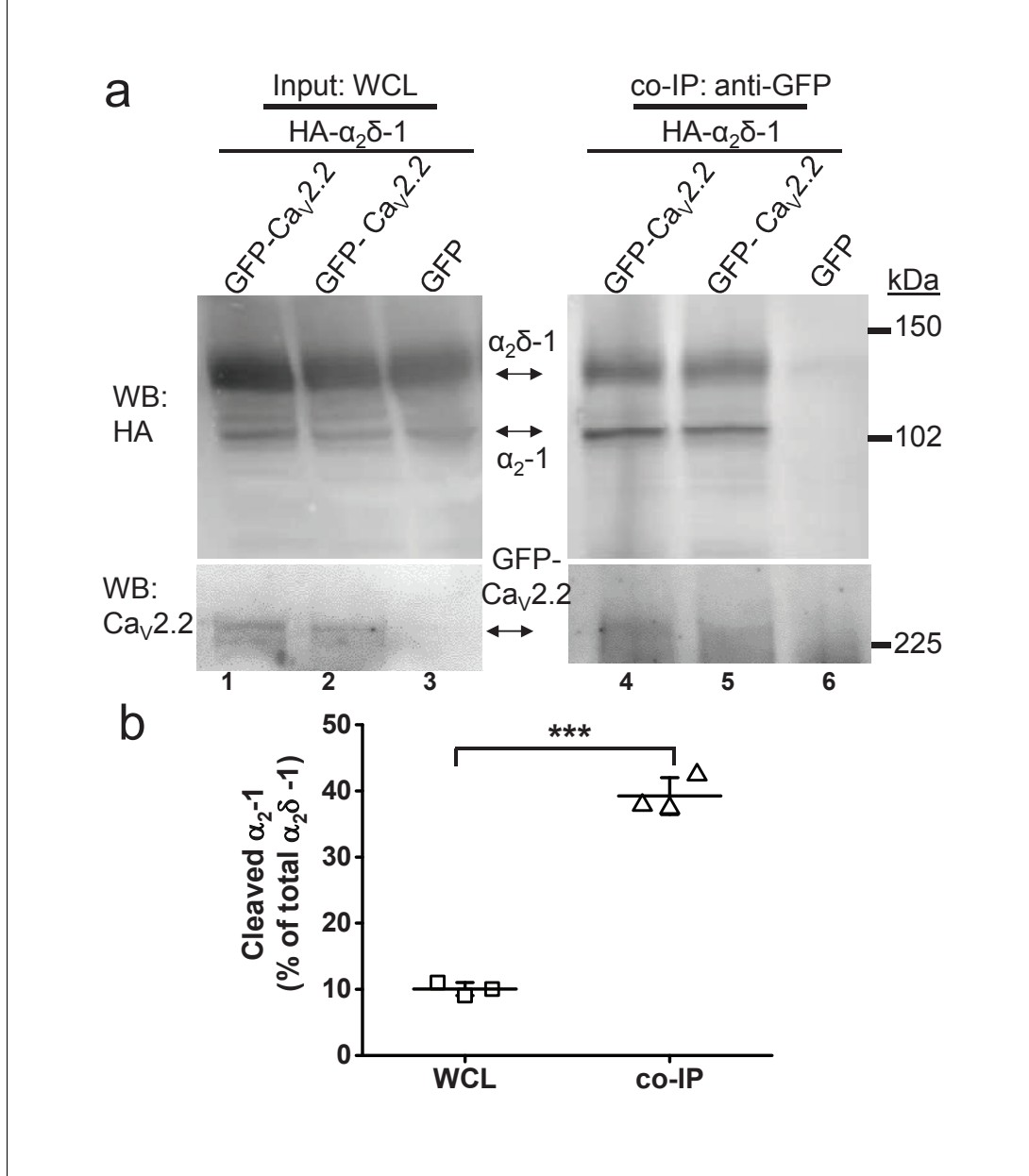

**Figure 4.** Quantified co-immunoprecipitation of Ca$_V$2.2 with cleaved and uncleaved fractions of wild-type α$_2$δ-1 from WCL of tsA-201 cells. (a) Left panels show WCL input from tsA-201 cells transfected with GFP-Ca$_V$2.2 (lanes 1 and 2) or GFP (lane 3), plus β1b and HA-tagged α$_2$δ-1: upper panel, HA-α$_2$δ-1 input; lower panel, Ca$_V$2.2-GFP input. Right panels show immunoprecipitation (IP) of GFP-Ca$_V$2.2 with anti-GFP Ab; immunoblots with Ca$_V$2.2 II-III loop Ab (lower panels, lanes 4 and 5) produced co-immunoprecipitation (co-IP) of HA-α$_2$δ-1 (corresponding upper panels lanes 4 and 5), revealed by anti-HA mAb. All samples deglycosylated. (b) Proteolytic cleavage of α$_2$δ-1 expressed as percentage of cleaved α$_2$-1 moiety to total α$_2$δ-1 calculated for input WCL fractions (squares) and for fractions co-immunoprecipitated with GFP- Ca$_V$2.2 (triangles). The cleaved α$_2$-1 moiety in the co-IP fractions is increased by 29.2 ± 1.7% compared with the WCL fractions (average of 3 independent experiments). ***p=0.0032, paired t-test.
DOI: https://doi.org/10.7554/eLife.37507.005

α$_2$δ and thrombospondin and its role in the mediation of synaptogenesis remains controversial (*Lana et al., 2016*; *Xu et al., 2010*), altogether these reports suggest that α$_2$δ subunits could play a role as an extracellular coordinator of synaptic function. Furthermore, the modulation of presynaptic Ca$_V$ channels by proteolytic cleavage of α$_2$δ subunits could serve as an additional regulatory mechanism for their complex synaptic functions at the post-translational level.

Ca$_V$2 channels and BK potassium channels are known to be part of multi-molecular complexes in neurons (*Berkefeld et al., 2006*; *Müller et al., 2010*). α$_2$δ−1 has very recently been shown to interact with BK channels, and this interaction was found to reduce the stability of Ca$_V$2.2 channels at the plasma membrane by preventing α$_2$δ−1 interacting with Ca$_V$2.2 channels (*Zhang et al., 2018*). Functionally, BK channels were shown to control neurotransmitter release by shortening the AP duration and reducing Ca$^{2+}$ influx into presynaptic elements at neuro-muscular junctions (*Protti and Uchitel, 1997*; *Yazejian et al., 1997*). Although their presence in presynaptic boutons has been disputed (*Hoppa et al., 2014*), BK channels are also expressed in axons from central neurons (*Debanne et al., 2011*; *Johnston et al., 2010*). Furthermore, α$_2$δ−1 has also recently been shown to interact with NMDA glutamate receptors (NMDARs) (*Chen et al., 2018*), albeit via a C-terminal domain of α$_2$δ−1 that is beyond the GPI-anchor attachment site and would therefore not be present in a mature GPI-anchored form (*Davies et al., 2010*; *Kadurin et al., 2012*; *Wu et al., 2016*). This interaction was found to promote the trafficking of the NMDARs to synaptic sites between peripheral dorsal root ganglion neurons and dorsal horn neurons in the spinal cord and is involved in the development of neuropathic pain (*Chen et al., 2018*). Therefore, it will be of great interest to determine whether fully mature α$_2$δ−1 is required for the interaction with BK potassium channels and with NMDARs.

Synchronous stimulated release is often followed by a delayed release occurring after the end of the stimulus, also called asynchronous release (*Atluri and Regehr, 1998*; *Goda and Stevens, 1994*; *Kaeser and Regehr, 2014*). Asynchronous release is thought to be activated by residual Ca$^{2+}$ remaining in the presynaptic terminal after the stimulation (*Atluri and Regehr, 1998*; *Cummings et al., 1996*). Although the source of Ca$^{2+}$ responsible for the initiation of synchronous release is indisputably identified from many studies as voltage-gated calcium channels within the active zone (*Catterall and Few, 2008*; *Dolphin, 2012*; *Nakamura et al., 2015*; *Zamponi et al., 2015*), the source of Ca$^{2+}$ involved in asynchronous release remains uncertain. To study asynchronous release in this work, we took advantage of the optical method developed previously (*Ariel and Ryan, 2010*) to monitor the slow increase of fluorescence of pHluorin tagged to vGlut-1 after the end of the high frequency stimulation (20 AP at 100 Hz). We show that asynchronous release is reduced in hippocampal presynaptic terminals when uncleaved α$_2$δ−1 (α$_2$(3C)δ−1) is expressed, and this inhibitory effect is abolished when 3C-protease is co-expressed. Together with our previous report showing that the proteolytic cleavage of α$_2$δ is critical for the functional trafficking of Ca$_V$2.2 channels to the presynaptic terminals (*Kadurin et al., 2016*), our data demonstrate that asynchronous release is mediated by Ca$^{2+}$ influx generated by Ca$_V$ localized at the presynaptic terminals. Relevant to our data, a study has characterized an asynchronous Ca$^{2+}$ current, recorded after the end of the stimulation pulse, conducted by both Ca$_V$2.1 and Ca$_V$2.2 channels and activated by the increase of intracellular Ca$^{2+}$ generated by the activity of these channels (*Few et al., 2012*). This asynchronous current was also identified in mouse hippocampal neurons and this led the authors to suggest that the asynchronous current could contribute to asynchronous release (*Few et al., 2012*). Other Ca$^{2+}$ sources for asynchronous release have been proposed (*Kaeser and Regehr, 2014*). Ca$^{2+}$-permeable P2X2 ATP receptors have been involved in asynchronous release in excitatory synapses between CA3 neurons and interneurons in the CA1 region in the hippocampus (*Khakh, 2009*). At these synapses, P2X2 receptors would be activated by ATP released from vesicles in presynaptic terminals. Further pharmacological characterization would be needed to ascertain the involvement of P2X2 receptors in the asynchronous release we are monitoring in our experimental model. Additionally, in the nucleus of the solitary tract, TRPV1 channels had been suggested to be a source of Ca$^{2+}$ for asynchronous release at excitatory synapses from unmyelinated cranial visceral primary afferent neurons (*Peters et al., 2010*). However, recent data from the same group have suggested instead that the Ca$^{2+}$ source for asynchronous release would originate from spill-over of intracellular Ca$^{2+}$ from Ca$^{2+}$ nanodomains created by Ca$_V$2 channels (*Fawley et al., 2016*). This latter hypothesis would fit well with our data showing that mature α$_2$δ−1 is needed to traffic Ca$_V$ to the presynaptic terminals to modulate asynchronous release.

Building on our previous report, we show here that the maturation of α$_2$δ is crucial for Ca$_V$ channels to fulfil their functional role on synaptic transmission. As α$_2$δ−1 expression is upregulated during chronic pain and increases presynaptic Ca$_V$2 trafficking (*Bauer et al., 2009*; *Kadurin et al., 2016*; *Patel et al., 2013*; *Zamponi et al., 2015*), α$_2$δ−1 represents a therapeutic target (*Zamponi, 2016*), and an important question to address for future studies will be to identify endogenous protease(s) involved in the proteolytic maturation of α$_2$δ proteins.

## Materials and methods

### Neuronal culture and transfection

All experiments were performed in accordance with the Home Office Animals (Scientific procedures) Act 1986, UK, using a Schedule one method. Hippocampal neurons were obtained from male P0 Sprague Dawley rat pups as previously described (*Hoppa et al., 2012*). Approximately $75 \times 10^3$ cells in 200 µl of plating medium (MEM (Thermo Fisher Scientific) supplemented with B27 (Thermo Fisher Scientific, 2%), glucose (Sigma, 5 mg/ml), transferrin (Millipore, 100 µg/ml), insulin (Sigma, 24 µg/ml), fetal bovine serum (Thermo Fisher Scientific, 10%), GlutaMax (Thermo Fisher Scientific,1%)) were seeded onto sterile poly-L-ornithine-coated glass coverslips. After 24 hr, the plating medium was replaced with feeding medium (MEM supplemented with B27 (2%), glucose (5 mg/ml), transferrin (100 µg/ml), insulin (24 µg/ml), Fetal bovine serum (5%), GlutaMax (1%) and cytosine arabinose (Sigma, 0.4 µM)) half of which was replaced every 7 days. At 7 days in vitro (DIV) and 2 hr before transfection, half of the medium was removed, and kept as 'conditioned' medium, and fresh medium was added. The hippocampal cell cultures were then transfected with mCherry, vGlut-pHluorin and either empty vector or $\alpha_2(3C)\delta-1$ or $\alpha_2(3C)\delta-1$ + 3C-protease (all cloned in pCAGGs) using Lipofectamine 2000 (Thermo Fisher scientific). After 2 hr, the transfection mixes were replaced with feeding medium consisting of 50% 'conditioned' and 50% fresh medium.

### Co-Immunoprecipitation

The protocol was adapted from a procedure described previously (*Gurnett et al., 1997*).

Briefly, a tsA-201 cell pellet derived from one confluent 75 cm$^2$ flask was resuspended in co-IP buffer (20 mM HEPES (pH 7.4), 300 mM NaCl, 1% Digitonin and PI), sonicated for 8 s at 20 kHz and rotated for 1 hr at 4°C. The samples were then diluted with an equal volume of 20 mM HEPES (pH 7.4), 300 mM NaCl with PI (to 0.5% final concentration of Digitonin), mixed by pipetting and centrifuged at 20,000 x g for 20 min. The supernatants were collected and assayed for total protein (Bradford assay; Biorad). 1 mg of total protein was adjusted to 2 mg/ml with co-IP buffer and incubated overnight at 4°C with anti-GFP polyclonal antibody (1:200; BD Biosciences). 30 µl A/G PLUS Agarose slurry (Santa Cruz) was added to each tube and further rotated for 2 hr at 4°C. The beads were then washed three times with co-IP buffer containing 0.2% Digitonin and deglycosylated as previously described alongside with aliquots of the initial WCL prior to co-IP. Laemmli buffer with 100 mM DTT was added to 1 x final concentration and samples were analysed by SDS-PAGE and western blotting with the indicated antibodies as described previously (*Kadurin et al., 2016*).

The human embryonic kidney tsA-201 cells were obtained from the European Collection of Authenticated Cell Cultures (# 96121229) and tested to be mycoplasma-free.

### Live cell imaging

Coverslips were mounted in a rapid-switching, laminar-flow perfusion and stimulation chamber (RC-21BRFS, Warner Instruments) on the stage of an epifluorescence microscope (Axiovert 200M, Zeiss). Live cell images were acquired with an Andor iXon+ (model DU-897U-CS0-BV) back-illuminated EMCCD camera using OptoMorph software (Cairn Research, UK). White and 470 nm LEDs served as light sources (Cairn Research, UK). Fluorescence excitation and collection was done through a Zeiss 40 × 1.3 NA Fluar objective using 450/50 nm excitation and 510/50 nm emission and 480 nm dichroic filters (for pHluorin) and a 572/35 nm excitation and low-pass 590 nm emission and 580 nm dichroic filters (for mCherry). Action potentials were evoked by passing 1 ms current pulses via platinum electrodes. Cells were perfused (0.5 ml min$^{-1}$) in a saline solution at 32°C containing (in mM) 119 NaCl, 2.5 KCl, 4 CaCl$_2$, 25 HEPES (buffered to pH 7.4), 30 glucose, 10 µM 6-cyano-7-nitroquinoxaline-2,3-dione (CNQX) and 50 µM D,L-2-amino-5-phosphonovaleric acid (AP5, Sigma). NH$_4$Cl application was done with this solution in which 50 mM NH$_4$Cl was substituted for 50 mM NaCl (buffered to pH 7.4). Images were acquired at 100 Hz over a 512 × 266 pixel area in frame transfer mode (exposure time 7 ms) and analyzed in ImageJ (http://rsb.info.nih.gov/ij) using a custom-written plugin (http://rsb.info.nih.gov/ij/plugins/time-series.html). Regions of interest (ROI, 2 µm diameter circles) were placed around synaptic boutons responding to an electrical stimulation of 100 AP at 10 Hz.

## Analysis

Data are given as mean ± SEM or as box (25–75%) and whiskers (10–90%) plots with mean and median (open squares and solid lines). Statistical comparisons were performed using unpaired Student's t test or one-way ANOVA with Bonferroni post-hoc test, using OriginPro 2016.

## Acknowledgements

This work was supported by a Wellcome Trust Investigator award to ACD (098360/Z/12/Z). We thank Dr. Matthew Gold for HRV-3C protease cDNA and Prof. Tim Ryan (Weill Cornell Medical College) for providing vGlut-pHluorin.

## Additional information

### Funding

| Funder | Grant reference number | Author |
|---|---|---|
| Wellcome | 098360/Z/12/Z | Annette C Dolphin |

The funders had no role in study design, data collection and interpretation, or the decision to submit the work for publication.

### Author contributions

Laurent Ferron, Conceptualization, Formal analysis, Investigation, Writing—original draft, Writing—review and editing; Ivan Kadurin, Conceptualization, Formal analysis, Investigation, Writing—review and editing; Annette C Dolphin, Conceptualization, Funding acquisition, Project administration, Writing—review and editing

### Author ORCIDs

Laurent Ferron  http://orcid.org/0000-0002-2547-7417
Annette C Dolphin  http://orcid.org/0000-0003-4626-4856

### Ethics

Animal experimentation: All experiments were performed in accordance with the Home Office Animals (Scientific procedures) Act 1986, UK, using a Schedule 1 method.

### Decision letter and Author response

Decision letter https://doi.org/10.7554/eLife.37507.010
Author response https://doi.org/10.7554/eLife.37507.011

## Additional files

### Supplementary files

• Supplementary file 1. Statistical information for *Figures 1–3*.
DOI: https://doi.org/10.7554/eLife.37507.006
• Transparent reporting form
DOI: https://doi.org/10.7554/eLife.37507.007

### Data availability

All data generated or analysed during this study are included in the manuscript.

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
