## [Decision Letter]

Thank you for submitting your article "Proteolytic Maturation of α_2_δ Controls the Probability of Synaptic Vesicular Release" for consideration by *eLife*. Your article has been reviewed by two peer reviewers, including Mary B Kennedy as the Reviewing Editor and Reviewer #1, and the evaluation has been overseen by Richard Aldrich as the Senior Editor.

The reviewers have discussed the reviews with one another and the Reviewing Editor has drafted this decision to help you prepare a revised submission.

Summary:

The study by Ferron et al. builds on an earlier publication in *eLife* from the Dolphin group showing that proteolytic processing of α_2_δ, an auxiliary subunit of N-type calcium channels (Ca_v_2.2) is necessary for its function in trafficking of the channel to the presynaptic terminal. Here they extend that study by showing that proteolytic processing increases the probability of release at presynaptic terminals, and modulates asynchronous release that follows a burst of action potentials. They state that this effect likely results from its effects on the function of presynaptic Ca_v_ channels. Preventing proteolytic processing by mutating the cleavage site results in a reduction of synaptic vesicle exocytosis as determined with pHluorin-tagged VGlut and reconstituting this processing by co-expressing a protease that cleaves the mutated cleavage site rescues the exocytosis. They also add the biochemical observation that proteolytic processing increases the strength of association of α_2_δ with the channel.

Minor revisions:

The study is appropriate as a research advance. The reviewers had some requested revisions that should be able to be accomplished quickly.

1) One reviewer noted that the use of optical methods to measure the RRP and asynchronous release is a relatively new technique. If the authors have used an additional more standard measure such as hyperosmotically induced exocytosis (high glucose), it would strengthen the paper to include it. Otherwise, the authors could briefly review the evidence supporting the validity of their assay in the Discussion.

2) References: The reference to Page et al., submitted, should be changed to "in press", or unpublished results, as appropriate. The Ariel and Ryan (2010) is important to validate the interpretation of the optical measurements of RRP and asynchronous release. The reference to Ariel and Ryan (2010) is incomplete and should include the article and doi: 4:18. doi: 10.3389/fncir.2010.00018.

3) Recent evidence indicates that α_2_δ also interacts with NMDARs. It would be good to discuss these findings in the Discussion in addition to the α_2_δ association with BK channels.

---

## [Author Response]

Minor revisions:The study is appropriate as a research advance. The reviewers had some requested revisions that should be able to be accomplished quickly.1) One reviewer noted that the use of optical methods to measure the RRP and asynchronous release is a relatively new technique. If the authors have used an additional more standard measure such as hyperosmotically induced exocytosis (high glucose), it would strengthen the paper to include it. Otherwise, the authors could briefly review the evidence supporting the validity of their assay in the Discussion.

The reviewer is right in saying that the use of this optical technique is relatively new to measure rapid exocytotic events, as it was first described in 2010 by Arial and Ryan (Ariel and Ryan 2010). However, several studies have now used it to investigate detailed mechanisms of synaptic vesicular release (Ariel et al., 2012, Hoppa et al., 2012, Jeans et al., 2017, Tagliatti et al., 2016). Moreover, this technique is listed as one of the 3 methods to measure RRP in a very recent review on "The Readily Releasable Pool of Synaptic Vesicles" by Kaeser and Regehr (Kaeser and Regehr 2017). Finally, Kyril E. Volynski's group (UCL) together with James E. Rothman's group (Yale University) are using this optical method to investigate the impact of mutation in synaptotagmin-1 on the RRP and asynchronous release in cultured mammalian neurons; unfortunately only an Abstract is so far published (Abstract Neuroscience 2017 – http://www.abstractsonline.com/pp8/#!/4376/presentation/28655), therefore we are unable to cite this.

We have now reviewed the evidence that validate the optical method we are using to estimate RRP and asynchronous release in the Discussion section of the manuscript.

2) References: The reference to Page et al., submitted, should be changed to "in press", or unpublished results, as appropriate. The Ariel and Ryan (2010) is important to validate the interpretation of the optical measurements of RRP and asynchronous release. The reference to Ariel and Ryan (2010) is incomplete and should include the article and doi:4:18. doi: 10.3389/fncir.2010.00018.

We have modified the reference Page et al. to unpublished observations and corrected Ariel and Ryan (2010).

3) Recent evidence indicates that α_2_δ also interacts with NMDARs. It would be good to discuss these findings in the Discussion in addition to the α_2_δ association with BK channels.

We have extended the Discussion to the newly described interaction between α_2_δ-1 and NMDA glutamate receptors (Chen et al., 2018).